# OpenReview forum: "HVT: A Comprehensive Vision Framework for Learning in Non-Euclidean Space"
_ICLR.cc/2025/Conference — ICLR 2025 Conference Withdrawn Submission_

### Official Review · Reviewer_Lshi · 2024-10-18

**Soundness:** 2
**Presentation:** 1
**Contribution:** 2
**Rating:** 3
**Confidence:** 5

**Summary:**

This paper investigates fully hyperbolic vision transformers. Based on advances in hyperbolic learning and fully hyperbolic convolutional approaches, this work takes the step towards transformer architectures. The work operates in the Poincaré ball and includes tricks such as additional learnable parameters in the positional embeddings and geodesic regularization. Experiments on ImageNet show that a hyperbolic ViT can outperform its Euclidean counterpart.

**Strengths:**

1)
The notion of fully hyperbolic vision transformers is an important step in hyperbolic learning. Based on advances in learning with hyperbolic embeddings at the end of a network, fully hyperbolic learning can open up entirely new doors in the field.

2)
An open question in fully hyperbolic learning is scalability, as also pointed out in this work. Yet the paper manages to scale its experiments to the ImageNet-level despite operating in the Poincaré ball. Moreover, this is done with good scores. Scaling to ImageNet is a positive sign for fully hyperbolic learning.

**Weaknesses:**

1+2)
The biggest issue with the paper is that most of the proposed components are not new. This limitation is coupled with extremely poor citation to existing literature. Many papers have in recent years investigated fully hyperbolic (visual) learning and have proposed neural network building blocks. Yet these are not cited.

The first contribution claimed in the introduction is that the authors proposed hyperbolic neural components. While they cite Hyperbolic Neural Networks by Ganea et al. (2018) in S2, they fail to mention this work throughout the method section. More importantly, Hyperbolic Neural Networks++ [a] is ignored. This paper not only outlines how to generalize MLPs and linear layers to the Poincaré ball, but even outline a self-attention algorithm in this space. Yet this manuscript does not mention the paper.

[a] Shimizu, Ryohei, Yusuke Mukuta, and Tatsuya Harada. "Hyperbolic neural networks++." ICLR (2021).

The related work on hyperbolic geometry in machine learning lists only 5 papers altogether and ignores recent work on fully hyperbolic convolutional learning, notably [b,c]:

[b] van Spengler, Max, Erwin Berkhout, and Pascal Mettes. "Poincare resnet." In Proceedings of the IEEE/CVF International Conference on Computer Vision, pp. 5419-5428. 2023.
[c] Bdeir, Ahmad, and Niels Landwehr. "Optimizing Curvature Learning for Robust Hyperbolic Deep Learning in Computer Vision." arXiv preprint arXiv:2405.13979 (2024).

In [b], a fully hyperbolic batch normalization is outlined, as well as linear layers, and more in the Poincaré ball. There are many more papers ignored, as summarized in [d,e]:

[d] Peng, Wei, Tuomas Varanka, Abdelrahman Mostafa, Henglin Shi, and Guoying Zhao. "Hyperbolic deep neural networks: A survey." IEEE Transactions on pattern analysis and machine intelligence 44, no. 12 (2021): 10023-10044.
[e] Mettes, Pascal, Mina Ghadimi Atigh, Martin Keller-Ressel, Jeffrey Gu, and Serena Yeung. "Hyperbolic deep learning in computer vision: A survey." International Journal of Computer Vision (2024): 1-25.

3)
The method outlines many small components but annoyingly never mentions which they claim as novel and which ones not. This is an issue since there is no citation in the method, except for one in the last sentence, making it impossible for the reader to understand which parts are copied from other papers and which ones are new. Below a summary of some issues:
- Hyperbolic linear layers have been proposed many times before, but not cited here.
- The ReLU and LayerNorm in Equation 10+11 are not hyperbolic, as the operations are performed in tangent space.
- The curvature in the positional embeddings (Eq 13) seems to just be a learnable parameter which has nothing to do with the curvature of the Poincaré ball.
- Self-attention in the Poincaré ball is given in [a].
- Residual connections in the Poincaré ball can be found in [b].
- Gradient clipping (Eq. 31) is common practice, see e.g. [f].
- Riemannian Adam (Eq. 32) is from Becigneul and Ganea (2019), although they are cited in Table 2.

As a result, the technical novelty of the paper is low, given that most components come from existing literature. This paper combines a number of them to create vision transformers.

[f] Dhingra, Bhuwan, Christopher J. Shallue, Mohammad Norouzi, Andrew M. Dai, and George E. Dahl. "Embedding text in hyperbolic spaces." arXiv preprint arXiv:1806.04313 (2018).

4)
The experiments are insufficient. Only 1 ImageNet result (Table 4) and 1 ablation study (Table 5) is given. Experiments on one dataset are however not enough. Moreover, multiple proposed components are never evaluated. In my view, the paper needs the following additional experiments to be more complete:
- Training and evaluation on other datasets. Can also be on smaller datasets.
- Starting from the pre-trained ImageNet model, fine-tune on smaller datasets and report test accuracy.
- Table 4 should also include runtime complexity and memory footprint. I expect that they will be much higher for HVT, but this is fine, it is good to be open about this.
- The geodesic regularization is not evaluated. What is its influence?
- Idem for layer scaling.

**Questions:**

I would like to see the rebuttal dive into the novelty of the paper and especially its relation to existing literature. Please outline what is considered new and what comes from other papers (with reference). Moreover, the weaknesses outline specific experiments that I would like to see discussed.

---

### Official Review · Reviewer_UbGP · 2024-10-29

**Soundness:** 2
**Presentation:** 3
**Contribution:** 2
**Rating:** 3
**Confidence:** 4

**Summary:**

The paper introduces the Hyperbolic Vision Transformer (HVT), aiming to enhance image classification in non-Euclidean space using hyperbolic geometry. By incorporating Möbius transformations and hyperbolic self-attention, the authors claim improvements in handling hierarchical data structures in images. Experiments on the ImageNet dataset show competitive performance gains over traditional Vision Transformers (ViTs) without substantial increases in model parameters.

**Strengths:**

The theoretical framework of the HVT model is well-developed, with detailed mathematical formulations. The paper also shows improvements on the ImageNet dataset.

**Weaknesses:**

My main concern is about the novelty. The paper relies heavily on established techniques and provides limited evidence of unique innovations beyond integrating hyperbolic geometry.

**Questions:**

As claimed in line 119-121, the main difference between this work and previous work is the domain focus. It looks like the experiments part didn’t show much on tasks beyond classification. Have the authors explored other vision tasks?

---

### Official Review · Reviewer_3VMA · 2024-11-02

**Soundness:** 3
**Presentation:** 3
**Contribution:** 2
**Rating:** 5
**Confidence:** 3

**Summary:**

The paper presents the Hyperbolic Vision Transformer (HVT), which extends the Vision Transformer (ViT) by incorporating hyperbolic geometry to model hierarchical relationships in image data more effectively. Key contributions include hyperbolic adaptations of positional embeddings, attention mechanisms, and linear layers, along with techniques for stability such as learnable curvature and gradient clipping. HVT outperforms standard ViTs on the ImageNet dataset, demonstrating its effectiveness in image classification without added model complexity.

**Strengths:**

It is well-executed, featuring detailed formulations and strong experimental validation. The clear presentation and effective adaptation of transformer components highlight its importance in advancing vision tasks that require complex relationship modeling.

**Weaknesses:**

It seems to me that this work, while effectively applying hyperbolic space knowledge to a standard computer vision task, primarily contributes through engineering improvements rather than deeply explaining why these adaptations lead to better performance. While the integration is valuable, emphasizing the underlying reasons for the performance gains would strengthen the contribution and provide more insight into its impact.

**Questions:**

The authors should include more analysis and explanations to clarify why the proposed method works. Providing deeper insights into the mechanisms behind the performance improvements would enhance the paper’s impact and help readers better understand the value of the approach.

---

### Official Review · Reviewer_Z2MD · 2024-11-03

**Soundness:** 2
**Presentation:** 3
**Contribution:** 2
**Rating:** 3
**Confidence:** 5

**Summary:**

In their paper, the authors create a hyperbolic transformer for the vision classification task. They rely on the Poincare model and create 1 to 1 translations of the Euclidean components. These can mainly be divided into 3 categories: the encoder, the attention calculation itself, and the final output layers. They perform an ablation study with the replaced components and show that a full hyperbolic model achieves the best performance. Additionally, they add a regularization term to the loss to ensure close distances between embeddings of the same class.

**Strengths:**

The paper clearly breaks down each of the operations and components present in the model. This makes the implementation very clear and helps reproducibility. Additionally, the method achieves good results on the proposed dataset.

**Weaknesses:**

There are some points of contention in both the methodological contribution and the claims made by the authors.

Beginning with the contributions, the authors do not present any novel components. They rely entirely on previously established methods for the linear layer, activations, hyperbolic attention calculation, and normalization.
As for the architecture, there is also no novelty. Fully hyperbolic vision models have been established before for both the Poincare and the Lorentz models [1] [2]. The papers also rely on hyperbolic normalization layers as opposed to normalization on the tangent space.

Finally, there are some questionable claims in the paper. The authors state in line 362 that the model is scalable, also in line 468, they restate this under the claim that the number of parameters is the same. However, this claim is not entirely true. Despite the similar number of parameters, hyperbolic models are extremely computationally inefficient. It would be nice to see the runtime and VRAM requirements compared to that of the Euclidean model. I would imagine this would be very prohibitive and it hinders scalability.

There are also other claims such as in line 94, " we utilize the Poincare ball model for its computational suitability in vision tasks". This is not explained and it is not referenced and I don't quite see what is meant by it vs the Lorentz model for example. Additionally, in line 140, they claim they are better vs previous hyperbolic models such as the Hyperformer but they never actually compare against them.

[1] van Spengler, Max, Erwin Berkhout, and Pascal Mettes. "Poincare resnet." Proceedings of the IEEE/CVF International Conference on Computer Vision. 2023.
[2] Bdeir, Ahmad, Kristian Schwethelm, and Niels Landwehr. "Fully Hyperbolic Convolutional Neural Networks for Computer Vision." arXiv preprint arXiv:2303.15919 (2023).

**Questions:**

The models were trained on 8 A100s. Was this with a batch size of 32 on each or a total batch size of 32?
Additionally, in line 471 you mentioned that Table 5 compares the impact of hyperbolic positional encoding, is this in the table or is it not updated?

---

### Official Review · Reviewer_rHjz · 2024-11-04

**Soundness:** 1
**Presentation:** 2
**Contribution:** 1
**Rating:** 3
**Confidence:** 4

**Summary:**

This paper proposes a fully-hyperbolic vision transformer (ViT) architecture. Experiments are performed on image classification with ImageNet dataset.

**Strengths:**

Proposes a new fully hyperbolic ViT architecture, including positional embeddings, attention mechanism, and feed-forward network, which shows some promising results compared to Euclidean ViT.

**Weaknesses:**

1. The paper claims theoretical analysis of their work, but this was not included in their paper. One suggestion for such an analysis would be to show that the Euclidean transformer block can be recovered in the limit $c \to 0$, where $c$ is the curvature of the Poincare ball.
2. The experiments are not robust, with only one experiment on one task and one dataset. It would be nice to see results on other image datasets, for example CIFAR-100.
3. The paper is not properly citing past work, such as for hyperbolic linear layers (e.g. [1]), making it difficult to tell which parts of the model are new.
4. I also think this paper would benefit from greater explanation of the motivation and intuition behind some of the methodsi n the methods section so the reader can better understand why certain choices were made. For example, why are the hyperbolic positional embeddings formulated as a Mobius scalar multiplication followed by a Mobius addition?

[1] Ganea, Octavian, Gary Bécigneul, and Thomas Hofmann. "Hyperbolic neural networks." Advances in neural information processing systems 31 (2018).

**Questions:**

1. Is it possible to compare against an Euclidean ViT with learnable scaling parameter $\beta$ in residual connections and head-specific scaling factors $\alpha_h$ (lines 351-353)?

---

### Note · Authors · 2024-11-12

I have read and agree with the venue's withdrawal policy on behalf of myself and my co-authors.